# A Depth-Enhanced Holographic Super Multi-View Display Based on Depth Segmentation

**DOI:** 10.3390/mi14091720

**Published:** 2023-08-31

**Authors:** Zi Wang, Yumeng Su, Yujian Pang, Qibin Feng, Guoqiang Lv

**Affiliations:** 1National Engineering Laboratory of Special Display Technology, National Key Laboratory of Advanced Display Technology, Academy of Photoelectric Technology, Hefei University of Technology, Hefei 230009, China; wangzi@hfut.edu.cn (Z.W.); 2022170016@mail.hfut.edu.cn (Y.S.); 2021170018@mail.hfut.edu.cn (Y.P.); fengqibin@hfut.edu.cn (Q.F.); 2School of Instrument Science and Opto-Electronics Engineering, Hefei University of Technology, Hefei 230009, China

**Keywords:** near-eye display, three-dimensional display, holographic display, super multi-view, depth of field

## Abstract

A super multi-view (SMV) near-eye display (NED) effectively provides depth cues for three-dimensional (3D) display by projecting multiple viewpoint or parallax images onto the retina simultaneously. Previous SMV NED have suffered from a limited depth of field (DOF) due to a fixed image plane. In this paper, a holographic SMV Maxwellian display based on depth segmentation is proposed to enhance the DOF. The proposed approach involves capturing a set of parallax images and their corresponding depth maps. According to the depth maps, the parallax images are segmented into N sub-parallax images at different depth ranges. These sub-parallax images are then projected onto N image-recording planes (IRPs) of the corresponding depth for hologram computation. The wavefront at each IRP is calculated by multiplying the sub-parallax images with the corresponding spherical wave phases. Then, they are propagated to the hologram plane and added together to form a DOF-enhanced hologram. The simulation and experimental results are obtained to validate the effectiveness of the proposed method in extending the DOF of the holographic SMV displays, while accurately preserving occlusion.

## 1. Introduction

Augmented reality (AR) technologies offer users an immersive three-dimensional (3D) experience, making them a crucial component in unlocking the full potential of the metaverse. Head-mounted displays (HMDs), also known as near-eye displays (NEDs), are popular visual interface devices used for AR due to their compact and portable design [1,2,3,4]. However, conventional NEDs often encounter the issue of vergence-accommodative (VAC) conflicts [5]. When viewing left and right images through corresponding eyes, the eyes rotate to observe three-dimensional images, and the vergence function accurately perceives depth based on the rotation information. However, the accommodation function is affected because the eyes focus on the image plane of the HMD imaging system instead of the actual 3D images. This discrepancy between vergence and accommodation leads to visual fatigue.

To overcome the VAC problem, several methods have been proposed. The integral imaging method [6,7,8,9,10] reproduces 3D images with an array of microlenses. The holographic NEDs try to provide the most natural depth cues by reconstructing the accurate optical wave field [11,12,13,14,15,16]. The multi-plane display [17,18,19,20,21] generates 3D images by displaying the 2D slices consisting of a 3D scene volume along the visual axis. The vari-focal plane method varies the distance of a single virtual image plane to different distances. While these methods provide good results, they are limited by the limited range of video sources and they are not very efficient. The Maxwellian display [22,23,24,25,26,27,28,29,30,31] partially alleviates the VAC problem by increasing the depth of field. However, it is unable provide accurate depth cues for monocular vision.

A super multi-view (SMV) display creates multiple viewpoints that are densely packed with an interval smaller than the eye’s pupil. It allows the rays from multiple viewpoints to be projected on the retina together, enabling the eye to focus on 3D images without the VAC problem. SMV NED’s video sources are more readily available because they are parallax images; it has a mature generation technique and a rich source of imagery. In addition, it has higher efficiency because the light only enters the pupil; there is no redundancy of 3D information.

In SMV displays, the DOF is a crucial parameter, which refers to the range within which the 3D images can be perceived clearly by the viewer. It is influenced by various factors, including the optical properties of the display system and the characteristics of the light rays emitted by individual pixels. By ensuring a wide DOF, the display system can accommodate the varying depth cues present in the 3D content. This helps to reduce visual discomfort and enhance the overall quality of the 3D images. The narrower the light ray is, the larger the DOF will be. Conventional SMV is based on geometric optics; however, it has a limited DOF [32,33,34]. To increase the DOF, an effective method is to limit the width of the light beam entering the pupil. Light-emitting diode (LED) light sources are usually used and collimated to illuminate the spatial light modulator (SLM) [35,36,37]. The finite size of the LED source influences the DOF of the system. In Ref. [38], an SMV Maxwellian display based on a collimated laser source was proposed where the limited light ray enhanced the DOF. The Maxwellian display converged the light ray into the pupil, which enlarges the DOF of the SMV display due to its small exit pupil. In ref. [39], a holographic SMV Maxwellian display was used to improve the DOF of the SMV display. This display offered the advantage of being compact and free from lens aberrations. It only required an SLM without the need for additional optical components. By leveraging the characteristics of Maxwellian displays, where a retinal image is still formed on the retina even when the eye is focused on the 3D image, the DOF of the SMV display can be expanded. However, it is important to note that Maxwellian displays do not possess an infinite DOF. They have a fixed virtual image plane, and the DOF range is typically centered around this image plane. As the 3D image moves away from the image plane, the image quality begins to deteriorate. Although numerical aperture filtering can be employed to limit the width of the light beam and increase the DOF, this approach may result in a loss of high-frequency components and a subsequent decrease in the image quality [40]. In other words, there exists a trade-off between the image quality and DOF.

In this paper, a holographic SMV Maxwellian display based on depth maps is proposed to enhance the DOF. The proposed approach involves capturing a set of parallax images and their corresponding depth maps. According to the depth maps, the parallax images are partitioned into N sub-parallax images at different depth ranges. Each sub-parallax image is then projected onto an image-recording plane (IRP) of the corresponding depth for hologram computation. In the hologram calculation process, the depth values of each sub-parallax are transformed into distances for Fresnel diffraction. The wavefront at each IRP is computed by multiplying the sub-parallax image with the corresponding spherical wave phase. These wavefronts are then propagated to the hologram plane and superimposed to generate the hologram that enhances the DOF. The simulation and experimental results are obtained to validate the effectiveness of the proposed method in extending the DOF of holographic SMV displays, while accurately preserving occlusion.

## 2. Conventional Holographic SMV Maxwellian Display with Limited Depth of Field

Figure 1 presents the principle of two or more viewpoints existing in the pupil to provide depth cues [32,33,34]. Figure 1a,b show that the human eye focuses on the display panel and a point of the 3D image, respectively. When the eye focuses on the display panel, the rays do not converge on the retina. In contrast, when the eye focuses on a point of the 3D image, as shown in Figure 1b, the rays converge on the retina. Thus, eyes can focus on 3D images, thereby preventing the VA conflict.

As depicted in Figure 2a, the previous holographic SMV Maxwellian display uses the implementation of multiple cameras to capture the three-dimensional scene, resulting in a set of parallax images [39]. This method was used to improve the DOF of the SMV display. In fact, the real scene could also be scanned with multiple cameras to obtain the depth information of the image in the way of the point cloud. In this experiment, two or more cameras with spacing *p* were used to capture the parallax images of the 3D scene while maintaining an identical field of view (FOV). To ensure that each object was clear, a camera with an infinite DOF was used. Following the capture process, Figure 2b illustrates the subsequent hologram generation process. At the same image-recording plane, each parallax image $I_{m}$(*x*,*y*) is multiplied with a specific convergent spherical wave phase and added together to form the complex amplitude distribution *U*(*x*,*y*):(1)$$U\left(x,y\right)=\sum_{m}^{M}I_{m}\left(x,y\right)\cdot \exp\left\{\frac{-jk\left[{\left(x-x_{m}\right)}^{2}+{\left(y-y_{m}\right)}^{2}\right]}{L}\right\}$$
where *k* = 2π/λ is the wave number, λ denotes the wavelength of the light, *L* is the distance from the IRP to the camera, and $z_{1}$ is the distance between the recording image plane and the hologram. Additionally, the position of the *m*-th viewpoint or camera is denoted as ($x_{m}$, $y_{m}$). The complex distribution at the hologram plane is obtained by a Fresnel diffraction based on a single fast Fourier transform:(2)$$H\left(x_{1},y_{1}\right)=\exp\left[\frac{jk\left(x_{1}{}^{2}+y_{1}{}^{2}\right)}{2z_{1}}\right]ℱ\left\{U\left(x,y\right)\exp\left[\frac{jk\left(x^{2}+y^{2}\right)}{2z_{1}}\right]\right\}$$

Then, this complex distribution can be encoded into an amplitude-type hologram by introducing a carrier wave:(3)A(x1,y1)=2Re[H(x1,y1)exp(jksinθy1)]+C
where *θ* represents the inclined angle of the carrier wave and *C* denotes a constant real value used to ensure a non-negative intensity distribution. It is important to note that the parallax images must be suitable for the hologram calculation, and this requires that the *FOV* of the camera satisfies certain conditions:(4)$$FOV=2\tan^{-1}\left[\frac{{\lambda z}_{1}}{2\mathrm{dL}}\right]$$
where d is the pixel pitch of the hologram.

Figure 3a,b shows the simulation and experimental results of a conventional holographic SMV Maxwellian display. A 3D scene was constructed using 3Ds Max 2012 modeling software, incorporating three objects positioned at different depths. Two cameras with an FOV of 6.49° were utilized to capture the parallax images. The distances of the three objects from the cameras were 400, 1000, and 1950 mm, respectively. We used an amplitude-type SLM (pixel pitch: 3.6 µm; resolution: 4096 × 2160, 180 Hz). Figure 3c presents the experimental system, where L is 400 mm, $z_{1}$ is 130 mm, and the pitch of the viewpoints p is 1 mm. In this experimental system, a collimated green laser beam passed through a polarization beam splitter (PBS) before illuminating the SLM. To eliminate interference fringes, we proposed a time-division method to destroy the temporal coherence of the laser source [39]. The SLM rapidly switched holograms to display these viewpoints, and these viewpoints entered the human eye at different times. The coherence length of the solid-state laser was approximately tens of meters; considering the speed of light, the coherence time of the laser was very short. As long as the time interval between two adjacent viewpoints is longer than the laser coherence time, they will not interfere with each other. However, due to the visual persistence, the brain perceives these two viewpoints simultaneously to form a 3D vision. In this experiment, we used spherical waves instead of the traditional random phase. Speckle is usually caused by an increase in the random phase, and this approach results in the better suppression of speckle and noise as well [28].

This reconstruction method effectively provided precise depth cues for monocular vision. Figure 3a,b present the results of the 3D-scene reconstruction at distances of 400, 1000, and 1950 mm. At each distance, only objects in focus were clear, while objects in other positions appeared out of focus and blurred. Notably, the simulation and experimental results exhibit the best image quality for object 1 when the focus is set at 400 mm. However, at reconstruction distances of 1000 and 1950 mm, the reconstruction quality of objects 2 and 3 noticeably deteriorated. This degradation can be attributed to the proximity of object 1 to the IRP compared to objects 2 and 3, which were situated at greater distances from the IRP. The limited DOF of the IRP significantly impacted the reconstructed image quality. In summary, the reconstruction of the holographic SMV display was accurate only around the IRP.

The underlying cause of the limited DOF can be observed in Figure 4. At the IRP, two identical pixels emit light rays towards each of the two viewpoints within the pupil. These light rays intersect in space, forming a real image point. The size of the image point *p* can be calculated as follows:(5)$$p=\frac{\Delta z}{\cos^{2}\beta }\alpha$$
where Δz is the distance between the image point and the IRP, *β* is the inclined angle, and α is the divergence angle of the light ray. It can be seen that, as Δz increases, the image point size increases, resulting in a degraded image quality. This is the reason for the limited DOF.

## 3. Depth-Enhanced Holographic SMV Maxwellian Display Based on Depth Segmentation

We proposed a multi-IRP method based on depth segmentation to enhance the DOF. In this method, each IRP recorded a portion of the 3D scene. As depicted in Figure 5a, in addition to capturing the parallax images, the cameras also need to capture corresponding depth maps. The depth map represents the change in depth of the 3D scene as a change in the gray scale. Depending on the depth value, the depth position of each point in the 3D scene can be determined. Quantify the continuous depth into N depth ranges [41,42,43]. Based on Equations (6) and (7), the parallax images are segmented into N sub-parallax images at different depth ranges and an IRP is created for each depth range:(6)$$I_{m}^{n}=I_{m}\cdot Mask^{n}(x,y)$$
where $I_{m}$ denotes the parallax image of the *m*-th viewpoint, $I_{m}^{n}$ denotes the *n*-th sub-parallax image of the *m*-th viewpoint, and the mask represents the process of segmenting the parallax image, which satisfies the following conditions:(7)$$Mask^{n}(x,y)=\{\begin{array}{c} 1\;if\;Depth(x,y)\in \left[\frac{255}{N}(n-1),\frac{255}{N}n\right]\\ 0\;\mathrm{else}\; \end{array}$$

Then, *N* sub-parallax images were projected onto *N* corresponding IRPs. As shown in Figure 5b, the overlapping planes FOV captured by the two cameras are defined as the Image capture plane (ICP) for ease of calculation. The distance from the ICP to the camera is $L_{\mathrm{ICP}}$. During the hologram calculation, the depth value of each sub-parallax image is converted to the corresponding Fresnel diffraction distance, and the wavefront at each IRP multiplies the parallax image by the corresponding spherical wave phase. Figure 5c shows that the complex hologram propagates the wavefront of each IRP to the hologram plane and adds them together:(8)$$H\left(x_{1},y_{1}\right)=\sum_{n}^{N}\mathrm{Pr}op\left\{\sum_{m}^{M}I_{m}^{n}\left(x,y\right)\cdot \exp\left\{\frac{-jk\left[{\left(x-x_{m}\right)}^{2}\right]+\left[{\left(y-y_{m}\right)}^{2}\right]}{L_{n}}\right\}\right\}$$
where $L_{n}$ is the depth value of *n*-th sub-parallax images, which is transformed into the Fresnel diffraction distance, and Prop{} represents the Fresnel propagation calculation.

During the projection process, the sub-parallax images on each IRP still have some mismatch problems, which may lead to inaccurate reconstructions. There are two influencing factors, and their solutions are proposed separately. The first problem is the position offset of the parallax images on the IRP. As shown in Figure 6a, since the IRP and the ICP are no longer in the same plane, the sub-parallax images do not exactly overlap on the IRP and must be adjusted and positioned for each set of sub-parallax images on each IRP. Taking the example of two viewpoints, as shown in Figure 6b, assume the offset of the sub-parallax images on the IRP as v. Based on the triangle relationship, we can derive the following equation:(9)$$v=\frac{\left(L_{ICP}-L_{n}\right)\cdot p}{2L_{\mathrm{ICP}}}$$

In order to facilitate calculations during the matching process, the resulting offset *v* was converted to an offset pixel value and corrected, which was then extended to multiple viewpoints.

As show in Figure 7, the second problem is that, in the IRP, the size of the Fresnel diffraction field does not match the size of the sub-parallax image obtained from the projection of the parallax image. During the Fresnel diffraction calculation, when using the FFT, the effective diffraction field size $S_{i}$ is limited by the pixel pitch of the SLM and calculated by:(10)$$S_{i}=N\cdot dx_{1}=\frac{{\lambda z}_{1}}{dx_{2}}$$
where *N* denotes the resolution of the SLM, $dx_{1}$ denotes the sample spacing on the IRP, $z_{1}$ is the distance from the SLM to the IRP, and $dx_{2}$ is the pixel pitch on the SLM. Let $S_{i}^{\prime }$ be the size of the sub-parallax image obtained from the projection of the parallax image according to the triangular relationship between the IRP and the ICP. The projection zone size is:(11)$$S_{i}^{\prime }=\frac{L_{n}\cdot S_{\mathrm{ICP}}}{L_{\mathrm{ICP}}}$$
where $S_{\mathrm{ICP}}$ is the size of the parallax image on the ICP. In order to reconstruct clear images, it is necessary to adjust the projection zone size until it completely overlaps with the Fresnel diffraction field size. The fitting process can be discussed in two ways. If ${\;S}_{i}<S_{i}^{\prime }$, the IRP size is used as the base, the overlap is retained, and the redundant part of the sub-parallax image is removed using the rectangular function. If ${\;S}_{i}>S_{i}^{\prime }$, the missing part is filled with the zero-padding function. The fitting process is as follows:(12)$$I^{\prime }=\{\begin{array}{c} rect\left(\frac{x}{S_{i}},\frac{y}{S_{i}}\right)\begin{array}{cc} & S_{i}<S_{i}^{\prime } \end{array}\\ Zeropadding\left(I\right)\begin{array}{ccc} & & S_{i}>S_{i}^{\prime } \end{array} \end{array}$$

Figure 8 shows the simulation and experimental results. Two viewpoints were created and simultaneously entered the pupil to project images onto the retina. Only three sub-parallax images were divided in the simulation and experiment for the sake of simplicity.

In this experiment, the maximum depth value $d_{\max}$ was 2000 mm and the minimum value $d_{\min}$ was 350 mm; the depth map was divided into three depth ranges. It can be seen that the three objects have equal reconstruction qualities from the experimental and simulation results. The reconstruction of objects 2 and 3 are better than the conventional method, confirming the enhanced DOF of the proposed method, and there is a clear occlusion relationship among the three objects in the front, middle, and back.

In order to quantitatively illustrate the enhancement of the proposed method for achieving multi-depth image quality, we performed PSNR calculations for both the simulation results of this method and the simulation results of the conventional method mentioned in Section 2, as shown in Figure 9. The conventional method [39] improved the DOF value of the SMV display using only the holographic SMV Maxwell display. However, the DOF of a single IRP was still limited. We constructed multiple image-recording planes based on the depth segmentation method, each with a DOF range, thus expanding the DOF. The image quality was significantly improved when compared with the conventional method. This method can also be extended to color, which requires the recording of information from all three RGB wavelengths into the same channel and the simultaneous use of a tri-color laser [24,44].

The proposed method in this paper is similar to the holographic stereogram (HS) [45,46,47,48], which also uses multi-view images to calculate the hologram. However, they have the following differences. First, the proposed method usually has a higher spatial resolution than the HS. In the HS, the hologram is usually divided into many hogels, each hogel recording the specific perspective information of the 3D scene. Thus, it has a trade-off between the spatial and angular resolutions, while the proposed method is non-hogel based, and multiple view images with full resolution are projected onto the eye. Second, the proposed method requires less information than the HS. In the HS, abundant parallax images from different viewpoints are obtained first and converted to hogels, while in the proposed method, only a few parallax images are captured corresponding to the viewpoints existing in the eye’s pupil. That is to say, the proposed method only needs to recover the light field information within the eye’s pupil.

## 4. Conclusions

A holographic SMV Maxwellian display based on depth segmentation was proposed to enhance the DOF. The difference from the traditional method was that we captured the depth map of the scene along with the parallax image, quantified the continuous depth into N depth ranges, and the complete parallax image was partitioned into N sub-parallax images at different depth ranges. These sub-parallax images were then projected onto the IRPs of the corresponding depths for the hologram computation. Through this projection process, this method created multiple IRPs to solve the problem of limited DOFs that exist. During the hologram calculation, the wavefront at each IRP was calculated by multiplying the sub-parallax images with the corresponding spherical wave phases. Then, they were propagated to the hologram plane and added together to form the DOF-enhanced hologram. This paper also solved the mismatch problem in the projection process by calculation and restored the accurate reconstruction. Simulations and experiments were conducted to validate the effectiveness of the method, demonstrating significant improvements in the DOF while preserving occlusion. The hologram calculation was simple and the video sources were easy to access and generate. The proposed method is promising for realizing holographic near-eye displays with a large DOFs.

## Figures and Tables

**Figure 1 micromachines-14-01720-f001:**
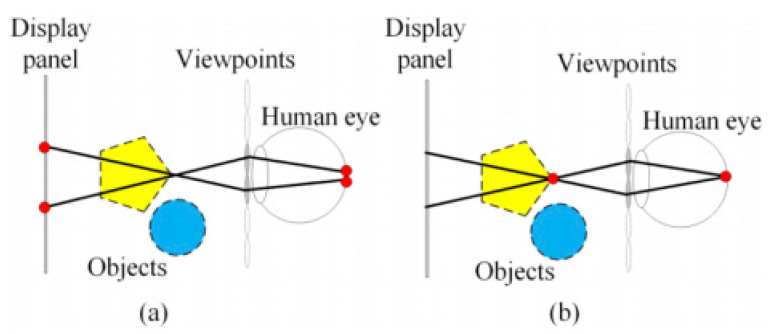
(**a**) Human eye focusing on the display panel. (**b**) Human eye focusing on a point on the 3D objects.

**Figure 2 micromachines-14-01720-f002:**
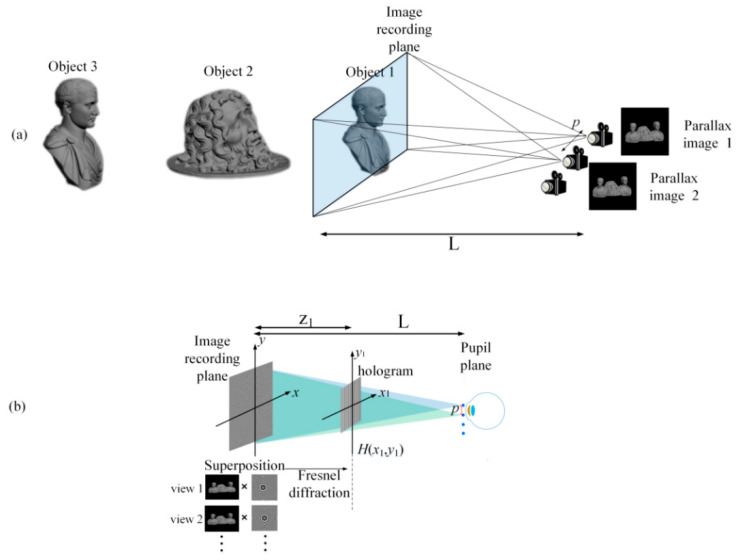
(**a**) Conventional parallax-capturing experiment (infinite DOF cameras are used to ensure that every object is clear); (**b**) conventional hologram generation with one IRP.

**Figure 3 micromachines-14-01720-f003:**
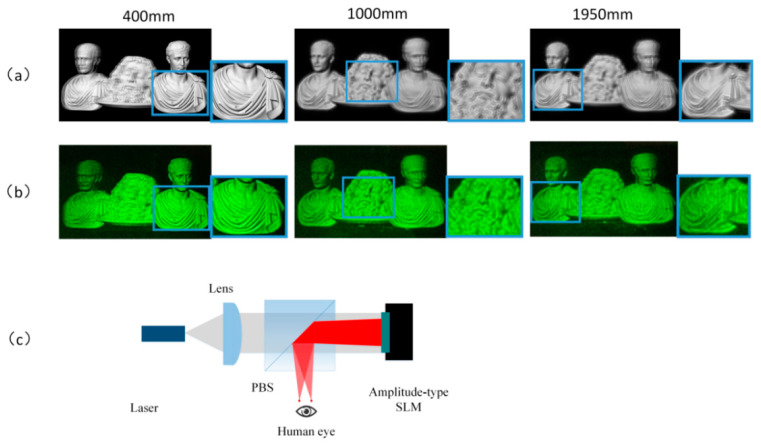
(**a**) Simulation in conventional holographic SMV display; (**b**) experimental results in conventional holographic SMV Maxwellian display; (**c**) experimental system.

**Figure 4 micromachines-14-01720-f004:**
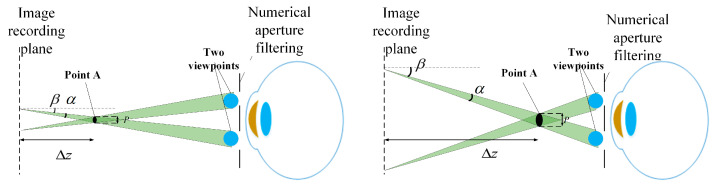
Principle of the limited DOF.

**Figure 5 micromachines-14-01720-f005:**
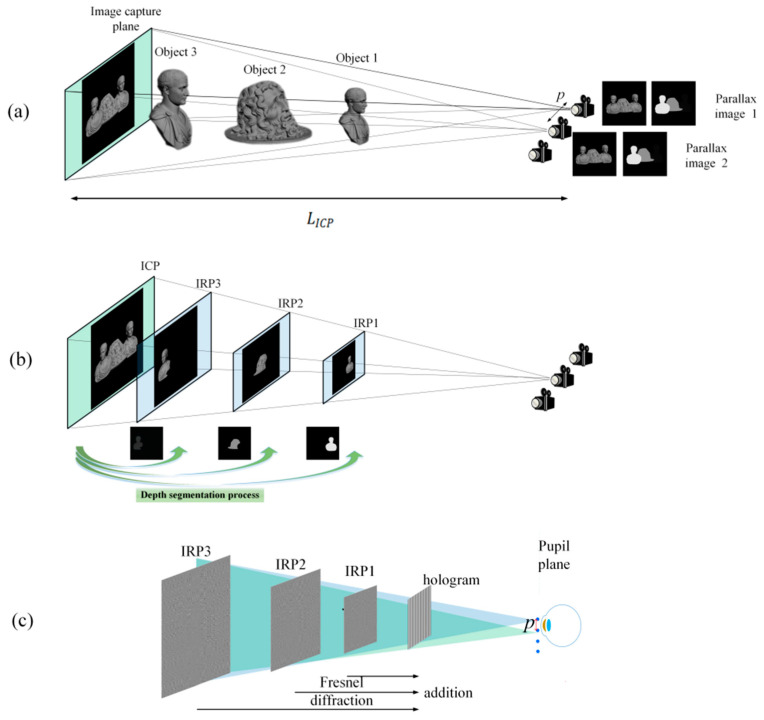
(**a**) The proposed process of capturing parallax and depth maps; (**b**) parallax image segmentation and projection process; (**c**) proposed hologram generation with multiple IRPs.

**Figure 6 micromachines-14-01720-f006:**
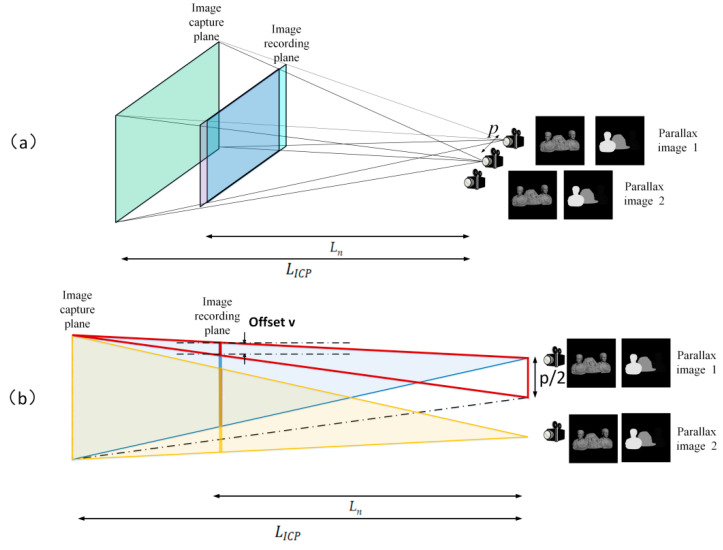
(**a**) Principle of non-overlapping sub-parallax images on the IRP. (**b**) Position offset of the parallax images on the IRP.

**Figure 7 micromachines-14-01720-f007:**
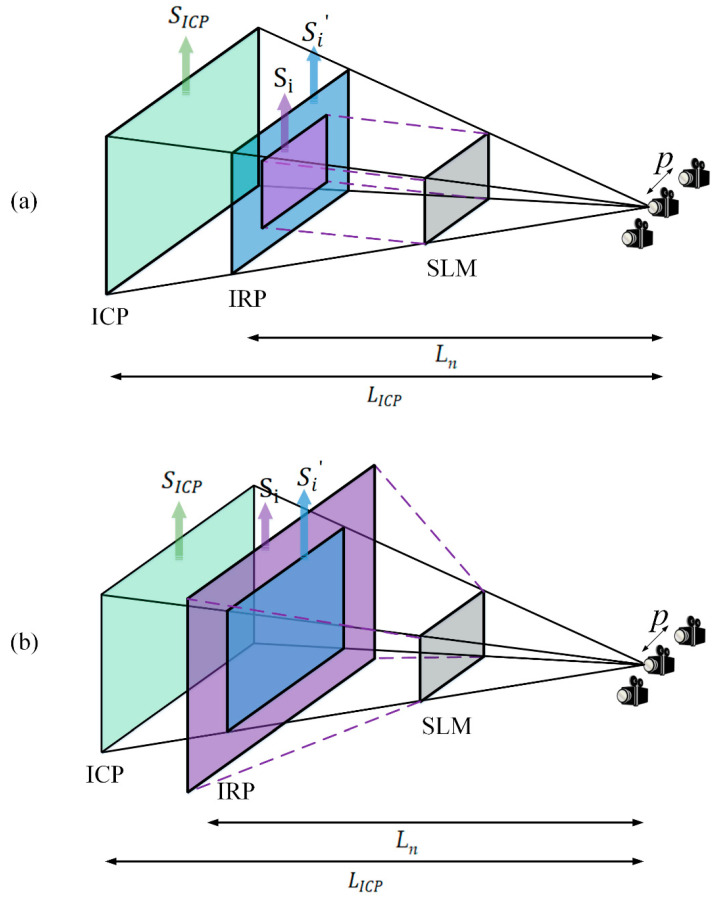
(**a**) When the diffraction field size is smaller than the projection zone size: $S_{i}<S_{i}^{\prime }$. (**b**) When the diffraction field size is larger than the projection zone size: ${\;S}_{i}>S_{i}^{\prime }$.

**Figure 8 micromachines-14-01720-f008:**
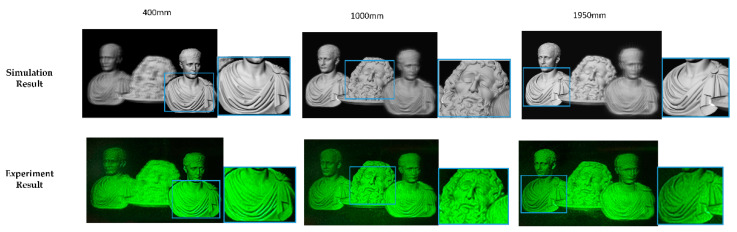
Simulation and experimental results of the proposed method.

**Figure 9 micromachines-14-01720-f009:**
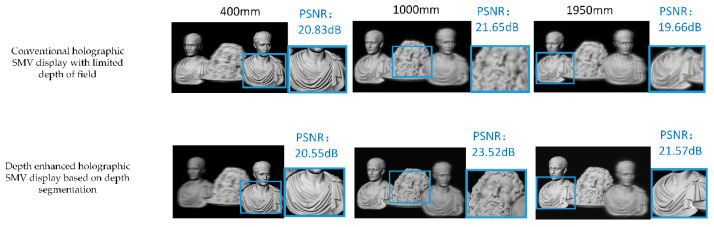
Comparison of PSNR of the proposed method with the conventional method.

## Data Availability

Data are contained within the article.

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
