# Peer review of "A Depth-Enhanced Holographic Super Multi-View Display Based on Depth Segmentation"

_micromachines, 2023, doi:10.3390/mi14091720_

Round 1
Reviewer 1 Report
This is an interesting paper which outlines a novel display system which has improved viewing characteristics and depth of field. The paper is mosty written well, but is at times a little confusing as it mixes terms such as holography and parallax which do not always mean the same thing in different planes. It is also not clear whether the display is either autostereoscopic or holographic and this is soemtimes difficult to follow in the description of the paper. There are a lot of papers being written which confuse holography with elements of Fresnel diffraction especially when considering near to eye applications. The depth enhanced method looks like a simplified layer-based CGH, each layer generates a hologram with corresponding depth and then they are added together, and this is why the quality is improved. Also, switching holograms of different viewpoints is a good way to display 3D vision. A few specific comments:
1) The system is described using Fresnel diffraction for enhance depth of focus. What type of illumination is being used in this display system? Are there any unwanted side-effects such as speckle?
2) Is the SLM being used in phase or amplitude mode? How does this effect the efficiency of the display? What is the frame rate of the SLM? Given it has a very high pixel density, how does this effect performance? Can it do colour? How does this effect the number of depth planes?
3) How are the depth maps determined? How realistic is this technique if the images do not contain discrete or individual planes? How does the optical design of the system proposed dictate the overall size of the near to eye display?
4) What effect does the defocused image have on the quality of the viewing experience? Is it a limit on performance?
5) The authors need to quantify the image improvement seen with their techniques. There are several metrics that can be applied to show improved performance.
The english reads pretty well, however there are a lot of acromyns used to describe the work which is a little confusing at times.
Author Response
请参阅附件

Reviewer 2 Report
This paper presents a method for converting a set of multiview plus depth pictures into holograms. It is described as a technique for enabling near eye displays with large depth of field since it provides a dense distribution of views that can be directed into a single eye.
The authors first describe a process for generating holograms from multiview data, that can be used in the context of maxwellian displays. Each captured image is multiplied with a zone plate associated to its optical center, and the resulting sub-holograms are propagated and summed into a single hologram.
From this (classical) method, that they present as the basis for their work, the authors identify the limitations in terms of depth of field and then introduce the main method that is claimed to solve these issues. Instead of only considering the pictures, they capture the depth of the scene and use it to partition the pictures into layers. Each layer for each viewpoint is then associated to a zone plate as in the background method, giving rise to a hologram that naturally renders the depth of field of the entire scene.
The paper is very well written and easy to read. The methods are exposed clearly and the figures perfectly complement the text. Whenever a specific issue is raised, it is well explained, along with the way to solve it.
However, to me this work suffers from a number of serious flaws:
- Lots of references are provided, that are globally significant, but do not represent the state-of-the art for the problem actually solved. Indeed, the works cited in the introduction are relevant since the purpose remains general, but as soon as more technical aspects are considered, the paper lacks both references and comparisions.
- The article is confusing regarding the use of the term "maxwellian", that usually refers to the fact that that the waves collapse at the center of the pupil (i.e. all rays pass through the center). It is not clear that the produced fields have this property, although I recognize that this ambiguity is present in several cited references (e.g. [34] and [35]). It seems to me that the output of the method is a general purpose hologram, with a depth of field corresponding to the depth of the captured scene, whereas with maxwellian displays there is no accommodation effort needed.
- The proposed method uses a layer representation and a propagation from these layers to the final hologram. Occlusions are handled using masks corresponding to the depth partitioning. This is far from being novel, and it is absolutely not clear what the method brings compared to similar works, e.g.
Antonin Gilles, Patrick Gioia, Rémi Cozot, and Luce Morin, "Hybrid approach for fast occlusion processing in computer-generated hologram calculation," Appl. Opt. 55, 5459-5470 (2016)
or
A. Gilles, P. Gioia, R. Cozot and L. Morin, "Computer generated hologram from Multiview-plus-Depth data considering specular reflections," 2016 IEEE International Conference on Multimedia & Expo Workshops (ICMEW), Seattle, WA, USA, 2016, pp. 1-6, doi: 10.1109/ICMEW.2016.7574699.
- Finally, I have a doubt regarding the adequacy to the scope of the journal, since the proposed work is hardly related to any device.
Author Response
请参阅附件

Reviewer 3 Report
In this manuscript, the authors propose a depth-enhanced holographic super multi-view display based on depth segmentation. The image misalignment issue is solved by careful analysis. Both simulation and experiment are discussed in enough detail, which is promising for near-eye 3D display. I recommend it for publication in Micromachines after the following questions and comments have been addressed satisfactorily.
1. Extra characters appeared in the upper and lower lines of some formulas, please delete.
2. The character “S” in formula 5 is not marked in Figure 3.
3. Suggest the use of PSNR to better confirm the quality improvement.
4. What is the diffrence between ICP and IRP?
5. The size relationship in Figure 6 is not obvious. It is recommended to use auxiliary lines.
Author Response
Please refer to the annex

Round 2
Reviewer 1 Report
Much better vesion
English is OK
Author Response
Replies are attached as annexes

Reviewer 2 Report
I aknowledge the efforts that have been made by the authors to clarify the issues that were raised. However, I was not conviced by the provided answers:
About the maxwellian aspect of the method: I understand that previous work has been made on a system that is qualified as maxwellian; in this approach, zone plates redirect the signal at the center of the pupil, which is enough to justify its maxwellian character. However, in the proposed method, the zone plates reorient to a dense grid of viewpoints, and hence the depth of field comes from the multiple views passing through the pupil. Hence this is not different from any "multiview to hologram" algorithm, be it adapted to near eye display.
About the comparision with existing methods: the comparision with "conventional holographic display with limited depth of field" is absolutely not convincing. The reference method is unknown and it is not clear what the reconstructions are (numerical? Optical?). Moreover, there should be a discussion on the reasons why the proposed method supposedly performs better.
Author Response
Replies are attached as annexes
